# LATENT POSTERIOR-MEAN RECTIFIED FLOW FOR HIGH-FIDELITY PERCEPTUAL FACE RESTORATION

## ABSTRACT

The perception-distortion (PD) tradeoff theory suggests that image restoration algorithms must balance perceptual quality and fidelity. To achieve minimal distortion while maintaining perfect perceptual quality, Posterior-Mean Rectified Flow (PMRF) proposes a flow-based approach where the source distribution is minimal-distortion estimates. Although PMRF is shown to be effective, its pixel-space modeling approach limits its ability to align with human perception. In this work, we propose **Latent-PMRF**, which reformulates PMRF in the latent space of a variational autoencoder (VAE), facilitating better alignment with human perception during optimization. By defining the source distribution on latent representations of minimal-distortion estimates, we bound the minimum distortion by the VAE's reconstruction error. Moreover, we reveal that the design of VAE is crucial, and our proposed **Sim-VAE** significantly outperforms existing VAEs in both reconstruction and restoration. Extensive experiments on blind face restoration demonstrate the superiority of Latent-PMRF, offering an improved PD-tradeoff compared to existing methods, along with remarkable convergence efficiency, achieving a $5.79\times$ speedup over PMRF in terms of FID. Our code will be publicly available.

## 1 INTRODUCTION

Face images are among the most common types of images, yet they often suffer from complex degradations during formation, recording, processing, and transmission (Wang et al., 2022a). Typical degradations, such as blur (Zhang et al., 2022), noise (Elad et al., 2023), downsampling (Dong et al., 2014; Luo et al., 2023; Liang et al., 2021), and JPEG compression (Jiang et al., 2021), can significantly degrade visual quality. Perceptual face restoration aims to recover high-quality, visually pleasing face images from degraded inputs. The key challenge lies in enhancing perceptual quality while maintaining fidelity. Recent studies show that generative models, particularly diffusion models (Sohl-Dickstein et al., 2015; Ho et al., 2020; Li et al., 2022; Wang et al., 2024) and flow matching models (Zhu et al., 2024; Ohayon et al., 2025), offer strong solutions for perceptual quality by modeling the distribution of natural images. Although such posterior modeling approaches can achieve perfect perceptual quality in theory, they do not guarantee minimal distortion under perfect perceptual quality constraints (Blau & Michaeli, 2018; Freirich et al., 2021; Ohayon et al., 2025). To minimize distortion, Posterior-Mean Rectified Flow (PMRF) (Ohayon et al., 2025) transports minimum distortion estimation to the target distribution using a rectified flow model. This approach can theoretically achieve minimal distortion (Freirich et al., 2021; Ohayon et al., 2025) under perfect perceptual quality constraints.

In this work, we challenge the necessity of constructing PMRF in the pixel space. While perceptual quality is formally defined as the statistical distance between the distributions of reconstructed and original images (Blau & Michaeli, 2018), researchers have found that distances in feature space better correlate with human perception (Heusel et al., 2017; Zhang et al., 2018; Szegedy et al., 2015; Sauer et al., 2021; Kumari et al., 2022). For instance, the most widely used metric for evaluating image generation models is the Fréchet Inception Distance (FID) (Heusel et al., 2017), which measuring distribution difference within the feature space of the InceptionNet (Szegedy et al., 2015). Additionally, many Generative Adversarial Networks (GANs) (Goodfellow et al., 2020) define discriminators in the feature spaces of pre-trained networks, such as EfficientNet (Sauer et al., 2021) and CLIP (Kumari et al., 2022). These findings suggest that measuring distribution distances in

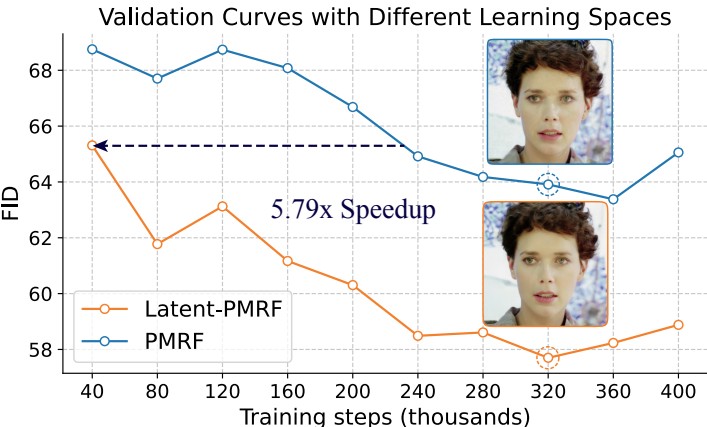

Figure 1: Illustration of perception optimization efficiency in latent space. We train PMRF and Latent-PMRF with the same compute budget. For VAEs with perceptual compression capabilities, differences in their latent space align better with human perception than those in pixel space, making latent space modeling more effective for perception optimization. Validation curves demonstrate the superior perceptual quality achieved by Latent-PMRF, with a $5.79\times$ speedup over PMRF in terms of FID.

feature space is an effective approach. Motivated by this, we propose reformulating PMRF in the latent space of a variational autoencoder (VAE) (Kingma, 2014), where perceptual quality can be optimized more efficiently, as shown in Figure 1.

While the idea appears straightforward, its optimality in terms of distortion requires careful analysis. Analogous to PMRF, we consider two distinct source distributions: (1) the posterior mean of latent representations, and (2) the latent representations of the posterior mean. We show that the second approach offers several advantages and is preferable. Most notably, it achieves minimal distortion bounded by the VAE's reconstruction error, which is not guaranteed by the first approach.

Overall, our **Latent-PMRF** can be understood as a rectified flow model (Liu et al., 2023b) in latent space, where the source distribution consists of the latent representations of the posterior mean and the target distribution consists of the latent representations of high-quality (HQ) images. While extensive research has explored latent space models for restoration tasks (Wang et al., 2024; Yue et al., 2023; Zhu et al., 2024; Lin et al., 2024; Gu et al., 2022; Liu et al., 2023a; Wang et al., 2022b), a fundamental question remains: are the commonly used VAEs sufficient for image restoration? We reveal that the VAEs employed in Stable Diffusion (SD) (Rombach et al., 2022), SDXL (Podell et al., 2024), and FLUX (Esser et al., 2024) are suboptimal for this task, as shown in Table 1. Unlike image generation, where increasing latent dimensionality often complicates optimization, restoration tasks benefit from a more informative latent space, as it reduces reconstruction error and thus lowers the minimal distortion bound.

To address this, we propose **Sim-VAE**, a simplified variant of SD-VAE, incorporating loss enhancements and architectural improvements that significantly improve both the VAE's reconstruction ability and the restoration performance of the final model. Our contributions are summarized as follows:

- Latent-PMRF achieves better alignment with human perception during optimization, resulting in a $5.79\times$ speedup over PMRF in terms of FID.

- The source distribution design of Latent-PMRF bounds the minimum distortion to the VAE's reconstruction error, and our improved Sim-VAE significantly boosts restoration performance when integrated with Latent-PMRF.

- Extensive experiments show that our Latent-PMRF achieves an improved PD-tradeoff and produces visually appealing results with high consistency to the inputs.

Table 1: **Comparison of VAEs** in CelebA-Test (Wang et al., 2021a). We evaluate the reconstruction performance of various VAEs and their effectiveness as latent spaces for Latent-PMRF. Notably, our Sim-VAE demonstrates significantly improved reconstruction capabilities and enhances the performance of Latent-PMRF in restoration. The best results are denoted in **bold**, and the second-best results are underlined. **f8c4** denotes $8 \times$ downsampling with 4 latent channels.

| VAE | Reconstruction | | | Restoration | | |
| --- | --- | --- | --- | --- | --- | --- |
| | PSNR↑ | LPIPS↓ | MMD$_{\text{DINOv2}}$ ↓ | PSNR↑ | LPIPS↓ | MMD$_{\text{DINOv2}}$ ↓ |
| SD1.5-VAE f8c4 | 30.463 | 0.044 | 0.6931 | 25.875 | 0.231 | 1.0302 |
| SD-XL-VAE f8c4 | 32.396 | 0.039 | 0.3973 | **26.481** | 0.263 | 1.0286 |
| FLUX-VAE f8c16 | 38.763 | 0.008 | 0.0675 | 26.152 | 0.245 | 1.0565 |
| SD-VAE f8c32 | 40.398 | 0.015 | 0.0986 | 25.265 | 0.222 | 0.8938 |
| **Sim-VAE** f8c16 | 37.903 | 0.026 | 0.0966 | 26.441 | **0.219** | 0.8918 |
| **Sim-VAE** f8c32 | **42.713** | **0.007** | **0.0511** | 26.382 | 0.224 | **0.8770** |

## 2 BACKGROUND

### 2.1 RECTIFIED FLOW

Rectified Flow (Liu et al., 2023b; Lipman et al., 2023; Albergo & Vanden-Eijnden, 2023) is a generative modeling approach that constructs a probability path $(p_t)_{0 \leq t \leq 1}$ from a source distribution $p_0$ to a target distribution $p_1$. Sampling involves drawing $X_0 \sim p_0$ and solving an Ordinary Differential Equation (ODE) defined by a velocity field $v_t$, which guides the transformation:

$$\frac{\mathrm{d}}{\mathrm{d}t} \psi_t(x) = v_t\left(\psi_t(x)\right), \quad \psi_0(x) = x. \tag{1}$$

The velocity field $v_t$ is parameterized by a neural network $v_t^\theta$ and trained via regression to match the conditional velocity field:

$$v_t\left(x_t \mid x_0, x_1\right) = x_1 - x_0, \tag{2}$$

where $X_t$ follows a linear interpolation between $X_0 \sim p_0$ and $X_1 \sim p_1$. The training objective is to minimize the Conditional Flow Matching (CFM) loss:

$$\mathcal{L}_{\text{CFM}}\left(\theta\right) = \mathbb{E}_{t, X_t, X_0, X_1} \left\| v_t^\theta(X_t) - (x_1 - x_0) \right\|^2. \tag{3}$$

### 2.2 POSTERIOR-MEAN RECTIFIED FLOW

Let $y$ denote a low-quality (LQ) image, which is a realization of a random vector $Y$ with probability density function $p_Y$, and let $x$ denote a high-quality image, which is a realization of a random vector $X$ with probability density function $p_X$. Posterior-Mean Rectified Flow (PMRF) is an image restoration framework designed to minimize distortion while preserving perceptual quality. PMRF achieves minimum distortion through two key stages:

1. **Posterior Mean Estimation**: A regression model is trained to estimate the posterior mean $x^* = \mathbb{E}[X|Y=y]$ given a LQ image $y$. This initial estimation step is theoretically optimal for minimizing the expected distortion between the predicted and true high-quality images.

2. **Rectified Flow**: Subsequently, a rectified flow model transforms the posterior mean estimation to match the true high-quality data distribution. This is achieved by learning a velocity field $v_t^\theta(\cdot)$ that guides the transformation through time $t$, enabling the model to recover fine details and natural variations present in the true data distribution.

The synergy between posterior mean estimation and flow-based modeling enables PMRF to achieve superior performance in image restoration tasks. By combining a distortion-optimal initial estimate with learned continuous transformations, PMRF successfully reconstructs high-fidelity images that are both perceptually pleasing and faithful to the original content.

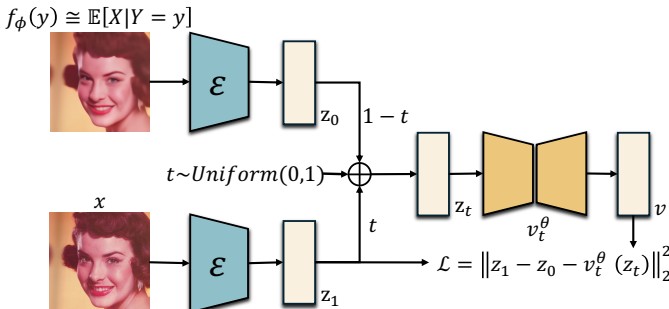

Figure 2: **Training Framework of Latent-PMRF.** We first estimate the posterior mean $\mathbb{E}[X|Y = y]$ from low-quality input $y$ using a pretrained estimator $f_\phi(\cdot)$. The posterior mean and the high-quality input $x$ are then encoded into latent representations $z_0$ and $z_1$. A flow network $v_t^\theta(\cdot)$ is trained to predict the velocity field along their linear interpolation: $z_t = (1-t)z_0 + tz_1$.

## 3 LATENT POSTERIOR-MEAN RECTIFIED FLOW

In this section, we introduce **Latent Posterior-Mean Rectified Flow (Latent-PMRF)**, a framework designed to achieve efficient perceptual quality optimization while preserving the distortion-minimizing property of PMRF. We first illustrate the Latent-PMRF framework and its implementation. Then, we provide the theoretical underpinnings, analyzing how our method addresses both perceptual quality and distortion. We prove that under ideal conditions, Latent-PMRF recovers the theoretical optimum of PMRF, and in practice, its performance is directly linked to the VAE's reconstruction fidelity.

### 3.1 THE LATENT-PMRF FRAMEWORK

The core idea of Latent-PMRF is to reframe image restoration as a transport problem in a learned latent space. Let $(\mathcal{E}, \mathcal{D})$ be the encoder and decoder of a pre-trained VAE, the latent representations of posterior mean $Z^* = \mathcal{E}(\mathbb{E}[X|Y])$ is defined as source distribution, and $Z = \mathcal{E}(X)$ is defined as target distribution. Then a rectified flow will be learned to transport $Z^*$ to $Z$ as shown in Figure 2. This simple framework is both efficient at perceptual quality optimization and able to achieve minimum distortion upper bounded by the reconstruction error of VAE, as illustrated in the following sections.

### 3.2 ANALYSIS OF PERCEPTUAL QUALITY AND DISTORTION

We now analyze how the Latent-PMRF design addresses the dual objectives of high perceptual quality and low distortion.

#### 3.2.1 EFFICIENT PERCEPTUAL QUALITY OPTIMIZATION

Operating in the latent space is particularly advantageous for optimizing perceptual quality, as supported by several established practices in the field. First, perceptual metrics like LPIPS Zhang et al. (2018), FID Heusel et al. (2017) typically measure differences in the feature space of pretrained networks. Second, GAN-based image generation methods Sauer et al. (2021); Kumari et al. (2022) successfully employ feature-space discriminators for improved visual quality. Furthermore, state-of-the-art diffusion models Rombach et al. (2022); Podell et al. (2024); Esser et al. (2024); Labs (2024) increasingly operate in VAE latent space, demonstrating the effectiveness of latent-space learning for perceptual quality optimization.

#### 3.2.2 THEORETICAL BOUNDS ON DISTORTION

While optimizing for perception, maintaining low distortion is paramount. Our choice of the posterior mean latent distribution $p(\mathcal{E}(\mathbb{E}[X|Y]))$ as the source is the key to achieving this. The posterior mean $\mathbb{E}[X|Y]$ is the optimal estimator under the MSE metric, establishing the theoretical lower bound on distortion. By initiating the transport from the latent representation of this optimal estimate, Latent-PMRF inherits this low-distortion property.

The connection to the theoretical optimum is formalized by the following theorems. Theorem 1 establishes this link under ideal conditions.

**Theorem 1** (Asymptotic Equivalence of Latent-PMRF and PMRF). *Let $(\mathcal{E}, \mathcal{D})$ be a VAE that forms an isometry between the data space and the latent space. Then, the optimal estimator $\hat{X}_{lat} = \mathcal{D}(T_{lat}(\mathcal{E}(X^*)))$ derived from Latent-PMRF is identical to the optimal estimator $\hat{X}_{PMRF}$ derived from PMRF, thus achieving the theoretical minimum distortion $D^* + W_2^2(p_{X^*}, p_X)$.*

*Proof.* The proof is provided in Appendix A.3. □

Theorem 1 shows that PMRF is a special case of Latent-PMRF under a perfect, isometric VAE. In practice, however, VAEs are imperfect and inevitably introduce reconstruction error. A natural question is how this imperfection affects the framework's total distortion. We show that the expected distortion decomposes into two terms: one inherent to the VAE's reconstruction error, and another governed by the transport process optimized by rectified flow. Consequently, reducing Latent-PMRF's distortion requires improving the VAE's reconstruction fidelity, which directly motivates our proposal of Sim-VAE. In addition, our framework can exploit pre-trained image-space estimators for the posterior mean, offering a practical advantage. See Appendix A.4 for details.

### 3.3 IMPROVED VARIATIONAL AUTOENCODER

For Latent-PMRF, the VAE not only sets the upper bound of restoration quality but also influences flow optimization. To this end, we introduce Sim-VAE, a streamlined variational autoencoder designed for high-fidelity reconstruction and better resolution generalization.

Sim-VAE departs from the widely used SD-VAE by: 1.**Simplifying ResBlocks** to remove redundant normalization/activation layers. 2.**Replacing group normalization with pixel-wise layer normalization** to avoid imbalanced feature activations. 3. **Eliminating self-attention in middle layers**, improving resolution generalization. 4. **Redistributing computation in resizing layers** by combining resolution and channel adjustments for greater efficiency. Full architectural details, comparisons, and training loss are provided in Appendix B.

## 4 EXPERIMENTS

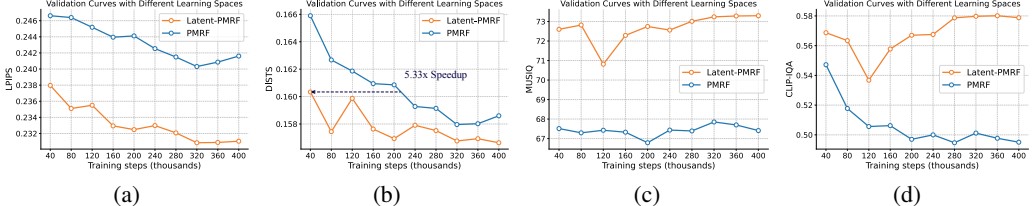

(a)                (b)                (c)                (d)

Figure 3: **Convergence Efficiency of Latent-PMRF.** We train both PMRF and Latent-PMRF using Sim-VAE for 400k iterations on FFHQ with a batch size of 64. Latent-PMRF significantly accelerates convergence, achieving a $5.33\times$ speedup in DISTS. It also outperforms PMRF in LPIPS, MUSIQ, and CLIP-IQA, achieving scores that PMRF cannot achieve within training. Furthermore, Latent-PMRF demonstrates strong performance even early in training, highlighting the importance of optimizing in a well-structured latent space.

### 4.1 EXPERIMENT SETUP

**Datasets.** We use two primary datasets: LSDIR (Li et al., 2023), containing 84,991 high-quality natural images, and FFHQ (Karras et al., 2019), which has 70,000 high-quality face images. For preprocessing, we crop LSDIR images into $512 \times 512$ patches and filter them using Q-Align (Wu et al., 2024) with a minimum score threshold of 3.5. FFHQ images are resized to $512 \times 512$.
**Implementation Details.** Sim-VAE is trained on a combination of the filtered LSDIR dataset and the first 10,000 images from FFHQ, using $256 \times 256$ image patches for 150,000 iterations with a batch size of 64. The Adam optimizer (Kingma & Ba, 2015) with default parameters and a cosine

Table 2: **Impact of VAE architectures** on CelebA-Test (Wang et al., 2021a). All VAEs use 32 channels. The results show that Sim-VAE significantly outperforms SD-VAE in both reconstruction and restoration tasks. Replacing $3 \times 3$ convolutions with self-attention causes training instability, making results unavailable.

| VAE | Reconstruction | | | Restoration | | |
|---|---|---|---|---|---|---|
| | PSNR↑ | LPIPS↓ | MMD$_{DINOv2}$ ↓ | PSNR↑ | LPIPS↓ | MMD$_{DINOv2}$ ↓ |
| **Sim-VAE** | 42.7129 | 0.0073 | **0.0511** | 26.3823 | 0.2236 | **0.8770** |
| - layernorm | **43.0518** | **0.0063** | 0.0619 | 26.1698 | 0.2270 | 0.8928 |
| - $3 \times 3$ conv | N/A | N/A | N/A | N/A | N/A | N/A |
| - interpolate | 42.9766 | 0.0075 | 0.0556 | 26.2465 | 0.2245 | 0.8817 |
| SD-VAE | 40.3979 | 0.0145 | 0.0986 | 25.2646 | **0.2224** | 0.8938 |

Table 3: **Impact of Latent Channels** on CelebA-Test (Wang et al., 2021a). Latent-PMRF benefits from richer latent representations, with 32 channels achieving a good balance across various metrics.

| Channel | Reconstruction | | | Restoration | | | |
|---|---|---|---|---|---|---|---|
| | PSNR↑ | LPIPS↓ | MMD$_{DINOv2}$ ↓ | PSNR↑ | LPIPS↓ | Q-Align↑ | MMD$_{DINOv2}$ ↓ |
| 16 | 37.9034 | 0.0261 | 0.0966 | 26.4412 | **0.2191** | 4.1006 | 0.8918 |
| 24 | 40.8142 | 0.0116 | 0.0603 | 26.3911 | 0.2251 | 4.1934 | **0.8657** |
| **32** | 42.7129 | 0.0073 | 0.0511 | 26.3823 | 0.2236 | 4.2934 | 0.8770 |
| 48 | **45.0554** | **0.0033** | **0.0485** | **26.4600** | 0.2264 | **4.3055** | 0.8863 |

learning rate schedule is used, decaying from $10^{-4}$ to $10^{-6}$ after a 500-step warm-up at $10^{-5}$. We set the latent channel to 32, unless specified otherwise.

Following PMRF, we utilize the posterior mean predictor trained by (Yue & Loy, 2024), and adopt HDiT (Crowson et al., 2024) as the velocity model of Latent-PMRF. The patch size is set to 1, and the transformer blocks are arranged as 2, 4, and 6 from high- to low-resolution. Depth-wise convolutions (Chollet, 2017) are incorporated into both the attention and feed-forward layers. Training is performed on FFHQ for 400,000 iterations with a batch size of 64. LQ images are synthesized following (Ohayon et al., 2025; Wang et al., 2021a). We use the Adam optimizer (Kingma & Ba, 2015) with $\beta_1 = 0.9$, $\beta_2 = 0.95$, and a fixed learning rate of $5 \times 10^{-4}$.

**Evaluation Metrics.** We evaluate our methods using a range of metrics grouped into four categories:

1. Reconstruction Fidelity: PSNR and MS-SSIM (Wang et al., 2003) assess reconstruction accuracy. For face restoration, we also include identity-related metrics like Deg (ArcFace embedding angle (Deng et al., 2019)) and landmark distance LMD (Yue & Loy, 2024).

2. Perceptual Similarity: LPIPS (Zhang et al., 2018) and DISTS (Ding et al., 2020) measure perceptual similarity between two images.

3. Non-Reference Metrics: CLIP-IQA (Wang et al., 2023), MUSIQ (Ke et al., 2021) and Q-Align (Wu et al., 2024) assess image quality without ground truth.

4. Statistical Distance: In addition to the commonly used FID (Heusel et al., 2017) for measuring distributional differences, we also consider FID$_{DINOv2}$ (Stein et al., 2024) and MMD$_{DINOv2}$ (Jayasumana et al., 2024). These metrics improve alignment with human perception using DINOv2 (Oquab et al., 2023) features, while MMD$_{DINOv2}$ further enhances sample efficiency using Maximum Mean Discrepancy (MMD) with an RBF kernel.

## 4.2 CONVERGENCE EFFICIENCY OF LATENT-PMRF

In this section, we demonstrate that constructing the PMRF in the latent space of Sim-VAE facilitates perception optimization, thus significantly accelerating convergence. As shown in Figure 1 and Figure 3, Latent-PMRF accelerates convergence by $5.79\times$ in terms of FID and $5.33\times$ in terms of DISTS. It also achieves significantly better LPIPS, MUSIQ, and CLIP-IQA scores, outperforming standard PMRF, which fails to reach similar performance within 400k training steps. The improved convergence efficiency of Latent-PMRF allows us to achieve strong results using relatively fewer computational resources during training.

## 4.3 IMPROVING LATENT-PMRF WITH BETTER VAE

**Effects of Architecture Design.** As illustrated in Section 3.3, we propose a series of architectural modifications aimed at improving the learning ability of the VAE and boosting restoration

Table 4: Quantitative comparisons on **CelebA-Test** (Wang et al., 2021a) benchmark. Our approach achieves the best PD-tradeoff, significantly reducing distortion while preserving top-tier perceptual quality. PMRF* denotes PMRF trained under the same compute budget as ours. Runtime is measured on NVIDIA A100. #Params (M) is reported as A + B, where A represents trainable parameters and B denotes frozen parameters.

| Method | PSNR↑ | MS-SSIM↑ | LPIPS↓ | DISTS↓ | Deg.↓ | LMD↓ | MUSIQ↑ | Q-Align↑ | FID↓ | FID$_{DINOv2}$↓ | MMD$_{DINOv2}$↓ | Runtime(s) | #Params(M) |
|---|---|---|---|---|---|---|---|---|---|---|---|---|---|
| GFP-GAN | 24.9861 | 0.8640 | 0.2407 | 0.1720 | 34.5372 | 2.4509 | 75.2940 | 4.7009 | 14.8021 | 223.0202 | 1.1638 | 0.0218 | 86.4 |
| RestoreFormer | 24.6157 | 0.8443 | 0.2416 | 0.1639 | 30.9218 | 1.9389 | 73.8584 | 4.5320 | 13.4083 | 152.1276 | 1.0003 | 0.0402 | 72.7 |
| CodeFormer | 25.1464 | 0.8589 | 0.2271 | 0.1700 | 35.7124 | 2.1389 | 75.5546 | 4.5835 | 15.3959 | 184.0517 | 1.1041 | 0.0349 | 94.1 |
| VQFR | 23.7626 | 0.8278 | 0.2391 | 0.1683 | 40.9100 | 3.0436 | 73.8407 | 4.5285 | 13.6547 | 199.7024 | 1.1287 | 0.0621 | 83.5 |
| DifFace | 24.7964 | 0.8233 | 0.2723 | 0.1679 | 44.1442 | 2.7230 | 69.0060 | 4.0769 | 13.5138 | 184.1844 | 1.0441 | 3.7054 | 159.7 + 15.7 |
| DiffBIR (v2) | 25.3946 | 0.8668 | 0.2654 | 0.1911 | 31.2931 | 1.5646 | 76.1659 | 4.8782 | 20.9181 | 156.9969 | 1.0692 | 6.3952 | 363.1 + 1319.3 |
| ResShift | 26.0359 | 0.8734 | 0.2464 | 0.1692 | 32.2866 | 1.8718 | 67.9784 | 4.2413 | 19.1850 | 167.3501 | 1.0534 | 0.6230 | 118.9 + 77.0 |
| FlowIE | 24.8349 | 0.8505 | 0.2312 | 0.1585 | 32.2254 | 1.7757 | 74.1167 | 4.6108 | 17.5334 | 164.6910 | 1.0733 | 0.3877 | 398.6 + 1319.3 |
| PMRF | 26.3321 | 0.8740 | 0.2232 | 0.1476 | 29.4504 | 1.5138 | 70.4967 | 4.2227 | 10.7225 | 96.8752 | 0.7214 | 0.5247 | 159.8 + 15.7 |
| PMRF* | 26.6431 | 0.8729 | 0.2407 | 0.1596 | 28.9294 | 1.3799 | 64.9143 | 3.7261 | 15.1663 | 140.6601 | 0.8578 | | |
| **Latent-PMRF** | **26.3887** | **0.8789** | **0.2207** | 0.1576 | 29.0961 | 1.5217 | 73.1496 | 4.3325 | 10.9447 | 110.4742 | 0.8108 | 0.5745 | 151.2 + 106.8 |

Figure 4: Qualitative comparisons on **CelebA-Test** (Wang et al., 2021a) benchmark. Our method produces visually appealing details while maintaining exceptionally high face identity preservation.

of Latent-PMRF. In this section, we demonstrate the practical implications of these modifications through controlled experiments. As shown in Table 2, we progressively remove various modifications to assess their impact on the reconstruction ability of the VAE and the restoration performance of Latent-PMRF trained on it. From the second row of the table, we observe that while replacing layer normalization with group normalization improves VAE fidelity, it degrades distributional faithfulness, and more importantly, severely hampers the restoration performance of Latent-PMRF. This suggests that group normalization negatively influences the learning of smooth features. The fourth row shows that using non-optimal resizing layers leads to poorer reconstruction and, consequently, worse restoration performance. Finally, when all modifications are removed, we obtain SD-VAE, which, while achieving good LPIPS in restoration, performs poorly in all other aspects.

**Impact of Latent Channels.** It is well known that increasing latent channels enhances the latent space representation and improves the VAE's reconstruction ability. However, the effect of latent channels on the restoration performance of Latent-PMRF remains unclear. As shown in Table 3, Latent-PMRF benefits from a richer latent space, with Q-Align scores consistently improving as the number of latent channels increases. We find that 32 channels strike a good balance across various metrics, so we set the default to 32.

## 4.4 COMPARISONS WITH STATE-OF-THE-ART METHODS

We primarily compare our method with PMRF (Ohayon et al., 2025), as our goal is to construct it in the latent space. Additionally, we compare with traditional approaches such as GFP-GAN (Wang et al., 2021a), RestoreFormer (Wang et al., 2022b), CodeFormer (Liu et al., 2023a), and VQFR (Gu et al., 2022), as well as recent diffusion-based methods like DifFace (Yue & Loy, 2024), ResShift (Yue et al., 2023; 2025), and DiffBIR (Lin et al., 2024). For a fair comparison, we reproduce ResShift using their official code but exclude the LPIPS loss used in their journal ver-

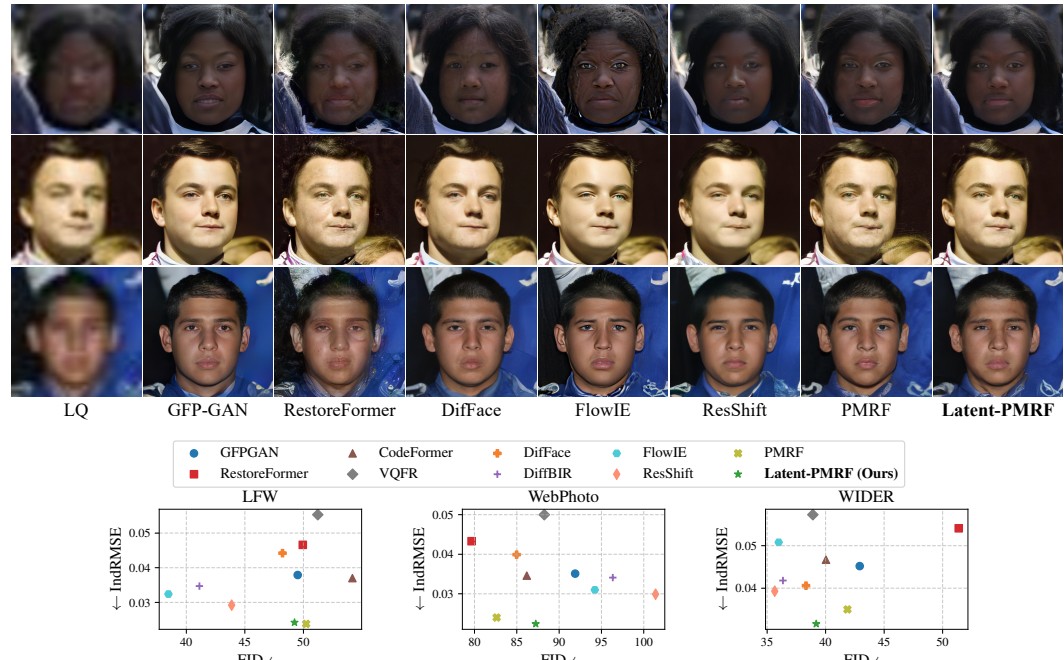

Figure 5: Comparisons on real-world datasets. Top: Qualitative results on the **WIDER-Test** (Zhou et al., 2022) dataset. Bottom: Comparison on the "distortion"-perception plane (IndRMSE vs. FID), where IndRMSE represents the RMSE of each method (Ohayon et al., 2025). Our method outperforms all others in IndRMSE, while achieving perceptual quality on par with the state-of-the-art.

sion. While incorporating this additional loss term is feasible, we omit it as it is not the focus of our work and requires computationally expensive VAE decoding during training. We also include FlowIE (Zhu et al., 2024), which also utilizes flow models. Notably, both DiffBIR and FlowIE leverage facial priors from large-scale Stable Diffusion (Rombach et al., 2022), whereas other methods use relatively smaller models.

**Results on Synthetic Dataset.** We evaluate our method on the CelebA-Test benchmark (Wang et al., 2021a). As shown in Table 4, PMRF and Latent-PMRF strike the best balance between distortion and perceptual quality. Specifically, only PMRF and Latent-PMRF achieve a PSNR above 26.3 dB and demonstrate superior face identity preservation, as evaluated by Deg. and LMD. In terms of statistical distance, PMRF, and our method learn more accurate distributions, outperforming others in FID, $FID_{DINOv2}$ and $MMD_{DINOv2}$. Notably, methods leveraging pretrained facial priors, such as GFP-GAN, DiffBIR, and FlowIE, achieve higher non-reference metric scores but tend to produce faces with lower faithfulness. In contrast, Latent-PMRF retains the high fidelity of PMRF while surpassing it in non-reference metrics. Moreover, Latent-PMRF demonstrates improved convergence properties—when the compute budget is reduced to match ours (scaling down from a batch size 256 and 3850 epochs (Ohayon et al., 2025)), PMRF experiences a significant performance drop. Overall, Latent-PMRF not only outperforms other methods but also converges much faster than PMRF.

We also present visual results in Figure 4. Compared to PMRF, our results generally exhibit better perceptual quality, which is reflected in the higher non-reference metrics we achieve. In contrast to other methods, which suffer from lower fidelity to the ground truth and consequently degrade face identity, our method preserves fine facial details while maintaining strong perceptual quality.

**Results on Real-world Datasets.** We evaluate the generalizability of Latent-PMRF on real-world datasets, including LFW (Huang et al., 2008; Wang et al., 2021a), WebPhoto (Wang et al., 2021a), and WIDER (Zhou et al., 2022). Since these datasets lack ground truth, we follow Ohayon et al. (2025) and use a pretrained posterior-mean estimator as a proxy for fidelity measurement. As shown in Figure 5, both Latent-PMRF and PMRF significantly outperform other methods in terms of fidelity, as indicated by IndRMSE. In terms of perceptual quality, Latent-PMRF outperforms PMRF on LFW and WIDER, while maintaining comparable performance to other methods. Overall, Latent-PMRF achieves a better perception-distortion tradeoff, offering comparable perceptual quality with superior distortion reduction. Visually, RestoreFormer produces poorly structured im-

ages, and FlowIE with the Stable Diffusion backbone shows artifacts with overly sharp details. In contrast, our method generates visually appealing images that remain consistent with the input.

## 5 RELATED WORK

**Generative Models in Latent Space.** Diffusion-based generative models (Ho et al., 2020; Dhariwal & Nichol, 2021) achieve impressive image synthesis but are computationally expensive, particularly for high-resolution images. Latent Diffusion (Rombach et al., 2022) mitigates this by learning distributions in a pretrained VAE's latent space, retaining only perceptually important information to enhance efficiency and scalability. Large-scale text-to-image models (Rombach et al., 2022; Podell et al., 2024; Esser et al., 2024) follow this paradigm, with VAE design playing a crucial role. Esser et al. (2024) show that increasing latent channels improves performance but requires larger generative models—for instance, even FLUX (12B parameters) (Labs, 2024) is limited to 16 latent channels. However, our Latent-PMRF framework greatly benefits from a more powerful VAE, since a stronger VAE enriches the source distribution with more information, thus alleviating the burden on the restoration process.

**Blind Face Restoration.** Blind face restoration aims to recover high-quality facial details from images degraded by unknown and complex factors while maintaining fidelity. From a training objective perspective, existing methods mainly fall into two categories: (1) GAN-based approaches (Wang et al., 2021a; 2022b; Gu et al., 2022; Liu et al., 2023a) optimize a weighted combination of distortion losses (e.g., L1, L2) and perceptual losses (e.g., adversarial loss (Goodfellow et al., 2020), perceptual loss (Johnson et al., 2016)), where the tradeoff between fidelity and perceptual quality is controlled by loss weighting (Blau & Michaeli, 2018; Ledig et al., 2017). (2) Posterior sampling-based methods (Lin et al., 2024; Zhu et al., 2024; Yue et al., 2023; Yue & Loy, 2024; Ohayon et al., 2025; Chen et al., 2024), particularly diffusion models, model the conditional posterior distribution of HQ images given degraded inputs. While these methods theoretically ensure superior perceptual quality, they often lead to suboptimal distortion (Ohayon et al., 2025).

PMRF (Ohayon et al., 2025) is the first approach to ensure optimal distortion under a perfect perceptual quality constraint. It first predicts the posterior mean (minimum distortion estimation) and then transports it to the HQ image distribution. However, we argue that distribution discrepancy in pixel space does not faithfully align with human perception. To address this, we propose constructing PMRF in the latent space of a VAE, which better optimizes perceptual quality. Furthermore, we design the source distribution to preserve PMRF's distortion-minimum properties in latent space.

**Concurrent works.** ELIR (Cohen et al., 2025) independently extends PMRF to the latent space of VAE. However, their focus is on improving testing-time efficiency via Consistency Flow Matching (Yang et al., 2024), while we aim to enhance optimization efficiency for perceptual quality. Furthermore, they use the posterior mean of latent representations as the source distribution, which, as discussed in Appendix A.2, is suboptimal. This choice leads to significant fidelity degradation in their model, whereas our Latent-PMRF preserves the high fidelity of PMRF.

## 6 CONCLUSION

We propose Latent-PMRF, which retains the minimal distortion property of PMRF while achieving better perceptual quality optimization. Our theoretical analysis shows that the latent representation of the posterior mean achieves a minimum distortion determined by the VAE's reconstruction error. Based on this insight, we introduce our Sim-VAE, with a series of modifications to enhance the reconstruction capability of the VAE, leading to a notable performance boost for Latent-PMRF. Latent-PMRF demonstrates remarkable convergence efficiency, achieving a $5.79\times$ speedup over PMRF in FID convergence. Furthermore, Latent-PMRF exhibits a better PD-tradeoff compared to existing methods in blind face restoration, with improved perceptual quality compared to PMRF. Although Latent-PMRF achieves strong performance, we observe a slight decrease in test speed compared to PMRF (see Table 4). This is because, while the velocity prediction in the latent space is faster, the encoding and decoding processes of the VAE are inherently slow. Improving the efficiency of the VAE could be a potential area for further enhancement.

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

## A  THEORETICAL ANALYSIS OF LATENT-PMRF

### A.1  PRELIMINARIES: THE DISTORTION-PERCEPTION TRADE-OFF

Our theoretical analysis is situated within the distortion-perception (DP) framework, using Mean Squared Error (MSE) as the distortion metric. We begin by defining the key random variables and the theoretical optimum for an ideal estimator.

- Let $X \sim p_X$ be the ground-truth image from the true data distribution.
- Let $Y$ be the corresponding degraded observation.
- Let $\hat{X}$ be an estimator for $X$, produced by a restoration model.
- Let $X^* = \mathbb{E}[X|Y]$ be the posterior mean estimate. $X^*$ is the optimal estimator of $X$ given $Y$ in the MSE sense, as it minimizes $\mathbb{E}[\|X - \hat{X}\|^2]$ over all possible functions of $Y$.

The total MSE distortion of any estimator $\hat{X}$ can be decomposed using the orthogonality principle of the posterior mean:

$$\mathbb{E}[\|X - \hat{X}\|^2] = \mathbb{E}[\|X - X^*\|^2] + \mathbb{E}[\|X^* - \hat{X}\|^2] \tag{4}$$

The first term, $D^* = \mathbb{E}[\|X - X^*\|^2]$, is the **irreducible error**. It represents the minimum possible distortion achievable by any estimator that only has access to the observation $Y$ and is independent of our specific choice of model for $\hat{X}$. The second term, $\mathbb{E}[\|X^* - \hat{X}\|^2]$, is the **optimizable error**, which our model aims to minimize.

We are interested in the case of **perfect perceptual quality**, where the distribution of the estimator's outputs must match the true data distribution, i.e., $p_{\hat{X}} = p_X$. As shown by Theorem 2 of Freirich et al. (2021), the problem of finding the minimum optimizable error under this constraint is equivalent to solving the optimal transport problem between the distributions $p_{X^*}$ and $p_X$. This minimum value is, by definition, the squared Wasserstein-2 distance, $W_2^2(p_{X^*}, p_X)$.

Therefore, the theoretical minimum total distortion for any estimator that satisfies the perfect perception constraint is given by:

$$D(0) = D^* + W_2^2(p_{X^*}, p_X) \tag{5}$$

This value serves as the gold standard against which we evaluate our Latent-PMRF framework under ideal conditions.

## A.2    DETERMINING OF SOURCE DISTRIBUTION

When extending PMRF into the latent space, two natural candidates for the source distribution arise:

1. **Mean of latent codes**: $Z_{opt1}^* = \mathbb{E}[\mathcal{E}(X)|Y = y]$
2. **Latent code of the mean**: $Z_{opt2}^* = \mathcal{E}(\mathbb{E}[X|Y]) = \mathcal{E}(X^*)$

In our Latent-PMRF framework, we adopt the latter, $Z^* = \mathcal{E}(X^*)$. The motivation is straightforward: decoding this choice yields the posterior mean $X^*$ up to the VAE's reconstruction error, thereby anchoring the transport process to the distortion-minimizing reference.

Specifically, the deviation introduced by Option 2 is simply

$$\|\mathcal{D}(\mathcal{E}(X^*)) - X^*\|^2,$$

which corresponds exactly to the **Inherent Reconstruction Error** in Theorem 2. Under a perfect isometric VAE (1), this error vanishes, and the source distribution coincides with the posterior mean.

By contrast, Option 1 introduces an additional, unavoidable error due to the nonlinearity of $\mathcal{D}$. Because

$$\mathcal{D}(\mathbb{E}[\mathcal{E}(X)|Y]) \neq \mathbb{E}[\mathcal{D}(\mathcal{E}(X))|Y],$$

even a perfect VAE would yield a non-zero deviation

$$\|\mathcal{D}(\mathbb{E}[\mathcal{E}(X)|Y]) - \mathbb{E}[X|Y]\|^2,$$

preventing the recovery of the theoretical optimum $D(0)$ guaranteed by Theorem 1.

Thus, choosing $Z^* = \mathcal{E}(X^*)$ cleanly isolates the VAE's reconstruction error as the *only* source of initial distortion—an error that is both theoretically principled and practically manageable. Moreover, this choice enables us to directly leverage powerful pre-trained estimators of the posterior mean in image space ($X^*$), eliminating the need to separately estimate latent posterior means.

## A.3    PROOF OF THEOREM 1: ASYMPTOTIC EQUIVALENCE

**Theorem 1** (Asymptotic Equivalence of Latent-PMRF and PMRF). *Let $(\mathcal{E}, \mathcal{D})$ be a VAE that forms an isometry between the data space and the latent space. Then, the optimal estimator $\hat{X}_{lat} = \mathcal{D}(T_{lat}(\mathcal{E}(X^*)))$ derived from Latent-PMRF is identical to the optimal estimator $\hat{X}_{PMRF}$ derived from PMRF, thus achieving the theoretical minimum distortion $D^* + W_2^2(p_{X^*}, p_X)$.*

*Proof.* The proof proceeds by establishing the theoretical minimum distortion under the perfect perception constraint and then showing that the Latent-PMRF estimator achieves this minimum. An isometry implies that the encoder $\mathcal{E}$ is a distance-preserving bijection with inverse $\mathcal{D}$, such that $\mathcal{D}(\mathcal{E}(x)) = x$ and $\|x_1 - x_2\| = \|\mathcal{E}(x_1) - \mathcal{E}(x_2)\|$. The Latent-PMRF estimator is defined as $\hat{X}_{lat} = \mathcal{D}(T_{lat}(\mathcal{E}(X^*)))$, where $T_{lat}$ is the optimal transport map between the latent distributions $p_{Z^*} = p_{\mathcal{E}(X^*)}$ and $p_Z = p_{\mathcal{E}(X)}$.

We first analyze the optimizable error term for this estimator, $\mathbb{E}[\|X^* - \hat{X}_{lat}\|^2]$. By applying the properties of the isometry, we can translate this pixel-space distortion into the equivalent transport cost in the latent space:

$$
\begin{aligned}
\mathbb{E}[\|X^* - \hat{X}_{lat}\|^2] &= \mathbb{E}[\|X^* - \mathcal{D}(T_{lat}(\mathcal{E}(X^*)))\|^2] \\
&= \mathbb{E}[\|\mathcal{D}(\mathcal{E}(X^*)) - \mathcal{D}(T_{lat}(\mathcal{E}(X^*)))\|^2] \quad &\text{(Since } \mathcal{D}(\mathcal{E}(X^*)) = X^*) \\
&= \mathbb{E}[\|\mathcal{E}(X^*) - T_{lat}(\mathcal{E}(X^*))\|^2] \quad &\text{(Since } \mathcal{D} \text{ is an isometry)} \quad (6) \\
&= \mathbb{E}[\|Z^* - T_{lat}(Z^*)\|^2] \\
&= W_2^2(p_{Z^*}, p_Z) \quad &\text{(By definition of } T_{lat})
\end{aligned}
$$

This is, by definition, the cost minimized by the optimal transport map $T_{lat}$ in the latent space, which equals $W_2^2(p_{Z^*}, p_Z)$. Furthermore, because the isometry preserves the cost function for transport, the Wasserstein distance itself is preserved between the spaces, meaning $W_2^2(p_{Z^*}, p_Z) = W_2^2(p_{X^*}, p_X)$.

Combining these results, the optimizable distortion of the Latent-PMRF estimator is equal to the theoretical minimum, $W_2^2(p_{X^*}, p_X)$. Therefore, the total distortion achieved by Latent-PMRF is $\mathbb{E}[\|X - \hat{X}_{lat}\|^2] = D^* + W_2^2(p_{X^*}, p_X) = D(0)$. This demonstrates that under the ideal condition of an isometric VAE, Latent-PMRF achieves the theoretical optimum distortion-perception trade-off.

$\square$

### A.4 PROOF OF THEOREM 2: THE INHERENT RECONSTRUCTION ERROR

**Theorem 2** (The Inherent Reconstruction Error in Latent-PMRF). *Let* $(\mathcal{E}, \mathcal{D})$ *be any VAE and* $\hat{X} = \mathcal{D}(T_{lat}(\mathcal{E}(X^*)))$ *be the estimator from Latent-PMRF. The total expected distortion (omit constant* $D^*$*) with respect to the posterior mean can be decomposed as:*

$$
\mathbb{E}[\|X^* - \hat{X}\|^2] = \underbrace{\mathbb{E}[\|X^* - \mathcal{D}(\mathcal{E}(X^*))\|^2]}_{\text{Inherent Reconstruction Error}} + \underbrace{\mathbb{E}[\|\mathcal{D}(\mathcal{E}(X^*)) - \hat{X}\|^2] + 2\mathbb{E}[\langle \ldots \rangle]}_{\text{Transport-Related Error}}
$$

*Proof.* The proof is based on an algebraic decomposition of the total error, which isolates the contribution of the VAE's reconstruction infidelity from the contribution of the transport process.

We define an intermediate term, $\tilde{X} = \mathcal{D}(\mathcal{E}(X^*))$, which represents the direct, non-transported reconstruction of the posterior mean. The optimizable error vector, $X^* - \hat{X}$, is then decomposed by adding and subtracting this term:

$$
X^* - \hat{X} = (X^* - \tilde{X}) + (\tilde{X} - \hat{X}) \tag{7}
$$

The vector $(X^* - \tilde{X})$ represents the error introduced solely by the VAE's imperfect reconstruction. The vector $(\tilde{X} - \hat{X})$ represents the change introduced by the transport process, viewed in the pixel space.

We compute the expected squared norm of the total error vector. Applying the identity $\|A + B\|^2 = \|A\|^2 + \|B\|^2 + 2\langle A, B \rangle$ and the linearity of expectation, we obtain:

$$
\begin{aligned}
\mathbb{E}[\|X^* - \hat{X}\|^2] &= \mathbb{E}[\|(X^* - \tilde{X}) + (\tilde{X} - \hat{X})\|^2] \\
&= \mathbb{E}[\|X^* - \tilde{X}\|^2] + \mathbb{E}[\|\tilde{X} - \hat{X}\|^2] + 2\mathbb{E}[\langle X^* - \tilde{X}, \tilde{X} - \hat{X} \rangle]
\end{aligned} \tag{8}
$$

Substituting the definitions of $\tilde{X}$ and $\hat{X}$ yields the expression in the theorem. This identity proves the decomposition. The first term, the **Inherent Reconstruction Error**, is non-negative and depends only on the VAE and the distribution $p(X^*)$; it is independent of the transport map $T$ and strictly positive for any imperfect VAE. The remaining terms constitute the **Transport-Related Error**, as they depend on the final estimator $\hat{X}$ and thus on the transport map $T$. This decomposition shows that the VAE's reconstruction fidelity imposes a fundamental component to the total distortion, which cannot be eliminated by the subsequent transport model. This concludes the proof. $\square$

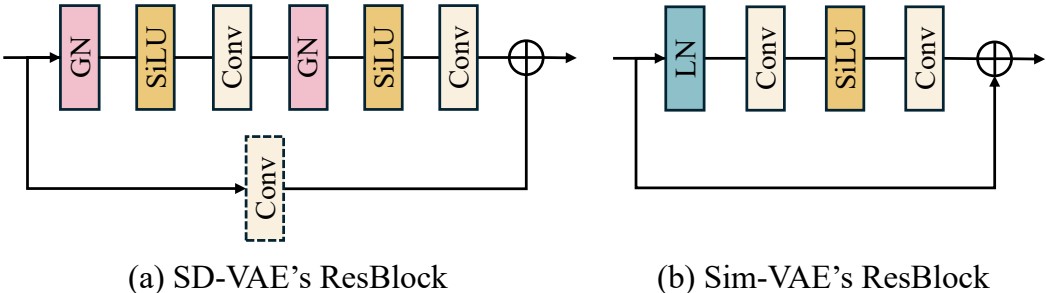



(a) SD-VAE's ResBlock     (b) Sim-VAE's ResBlock



Figure 6: Comparison of ResBlock designs between SD-VAE and Sim-VAE. Sim-VAE simplifies the ResBlock architecture by removing redundant components.

## B   IMPROVED VARIATIONAL AUTOENCODER

In this section, we detail the design of **Sim-VAE**. For the Latent-PMRF model, the VAE not only defines the upper bound for restoration performance but also affects the optimization of flow model. We first outline several architectural improvements aimed at enhancing both the reconstruction ability of the VAE and the distortion lower bound of Latent-PMRF. Next, We overview our training loss, where we propose eliminating the adversarial loss when VAE is strong enough, simplifying the training procedure.

### B.1   ARCHITECTURE IMPROVEMENTS

Our VAE architecture builds upon the classical VQGAN Esser et al. (2021), which has been widely adopted in numerous influential works Rombach et al. (2022); Podell et al. (2024); Esser et al. (2024); Labs (2024). We refer to this architecture as SD-VAE, reflecting its widespread adoption since Stable Diffusion. The encoder and decoder share a symmetric architecture, so we focus on describing the encoder, as the decoder follows an analogous structure in reverse.

**Simplified ResBlock:** Inspired by recent efficient convnet designs Liu et al. (2022); Chen et al. (2022), we propose a simplified ResBlock He et al. (2016) (Figure 6) that uses only one activation function and one normalization layer per block, improving efficiency without sacrificing performance.

**Pixel-wise Layer Norm:** The SD-VAE has been shown to produce imbalanced feature representations, where certain regions in intermediate feature maps exhibiting disproportionately high magnitudes Sadat et al. (2024); drhead (2024), as illustrated in Figure 8. While these local outliers in the feature maps serve to preserve global information drhead (2024), they may complicate latent diffusion model training. Inspired by Sadat et al. (2024); Karras et al. (2020), we propose replacing group normalization Wu & He (2018) with pixel-wise layer normalization Lei Ba et al. (2016); Chen et al. (2022), which normalizes each spatial location independently and promotes more balanced feature representations.

**Removing Self-Attention in Middle Layers:** SD-VAE uses self-attention Vaswani et al. (2017) in middle layers to capture global context, but this introduces a key limitation: resolution generalization issues. VAEs are usually trained on fixed low-resolution inputs, and global operators like self-attention often struggle to maintain performance across different resolutions during inference Press et al. (2022); Gao et al. (2025). While fine-tuning on high-resolution data is a common solution Sadat et al. (2024); Chen et al. (2025), it complicates training with additional optimization stages. To address this, we propose a simple modification: replacing self-attention with standard

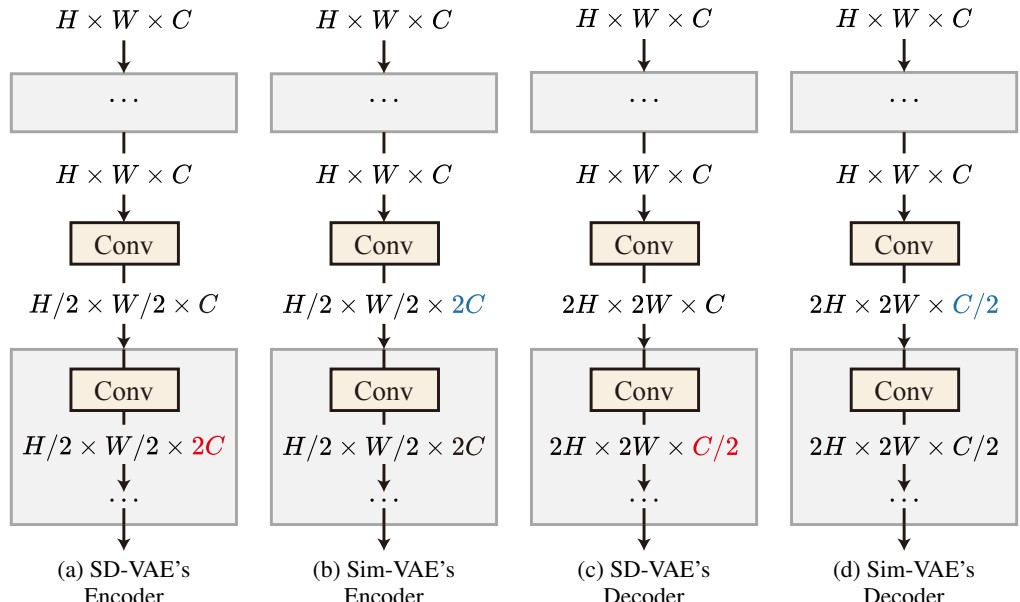

|  |  |  |  |
|---|---|---|---|
| (a) SD-VAE's Encoder | (b) Sim-VAE's Encoder | (c) SD-VAE's Decoder | (d) Sim-VAE's Decoder |

Figure 7: Illustration of the resizing layer design. Sim-VAE redistributes computation to ensure that channel dimension adjustments occur immediately with resolution changes.

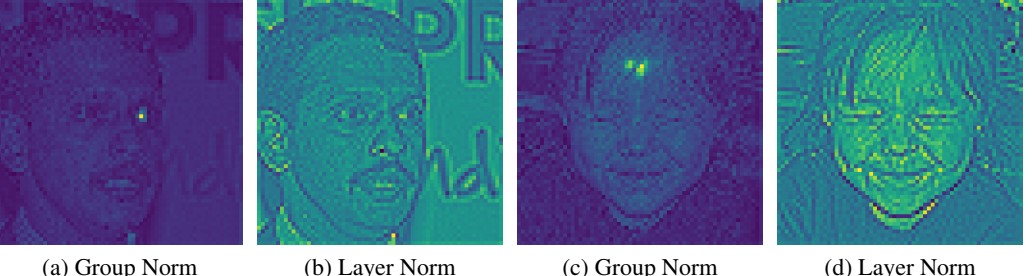

|  |  |  |  |
|---|---|---|---|
| (a) Group Norm | (b) Layer Norm | (c) Group Norm | (d) Layer Norm |

Figure 8: Two examples of the latent representations. Using pixel-wise layer normalization instead of group normalization allows the model to learn more balanced feature maps.

$3 \times 3$ convolutional layers, which offer better generalization across different resolutions.

**Redistribute Parameters between Resizing Layers:** In SD-VAE, resizing layers are responsible handling stage transitions, but the original design separates resolution changes from channel adjustments (Figure 7a): resizing layers maintain channel dimensions, while later convolutional layer handle channel modifications. This creates bottlenecks during downsampling and retains inefficiently high-dimensional features during upsampling. We propose integrating channel adjustments directly into the resizing layers—expanding channels during downsampling and reducing them during upsampling. This change improves information preservation and computational efficiency without increasing parameter count or complexity.

### B.2 TRAINING LOSS

The training objective for autoencoders typically comprises three components Esser et al. (2021): a reconstruction loss $\mathcal{L}_{\text{recon}}(\mathcal{D}(\mathcal{E}(x)), x)$ that measures the similarity between input and reconstructed images, a regularization term $\mathcal{L}_{\text{reg}}(\mathcal{E}(x))$ that constrains the latent space, and an adversarial loss Goodfellow et al. (2020) $\mathcal{L}_{\text{adv}}$ that encourages photorealistic reconstructions by discriminating between real images $x$ and their reconstructions $\mathcal{D}(\mathcal{E}(x))$. We observe that with sufficient model capacity, the adversarial loss becomes unnecessary without compromising performance, as also observed in Qwen-Image Wu et al. (2025). Thus, the training loss simplifies to:

$$\mathcal{L}_{\text{train}} = \mathcal{L}_{\text{recon}} + \lambda_{\text{reg}} \mathcal{L}_{\text{reg}} \tag{9}$$

The reconstruction loss $\mathcal{L}_{\text{recon}}$ combines $\ell_1$ distance with perceptual loss Johnson et al. (2016), following the weighting scheme of Real-ESRGAN Wang et al. (2021b). For regularization, we use the Kullback-Leibler (KL) divergence as $\mathcal{L}_{\text{reg}}$, with $\lambda_{\text{reg}}$ set to $10^{-6}$ as in Rombach et al. (2022).

## C    FURTHER EXPERIMENTS

To further validate the convergence of our Sim-VAE, we directly compare latent-PMRF (based on FLUX-VAE) and and Latent-PMRF (based on Sim-VAE). As shown in Figure 9, even with the same number of latent channels (aligned computational cost), Sim-VAE exhibits superior convergence property.

In addition, we conduct comparative experiments on real-world datasets. As shown in Figure 10, Latent-PMRF achieves significantly better results than PMRF, while attaining comparable MUSIQ scores and substantially improved IndRMSE compared to state-of-the-art methods.

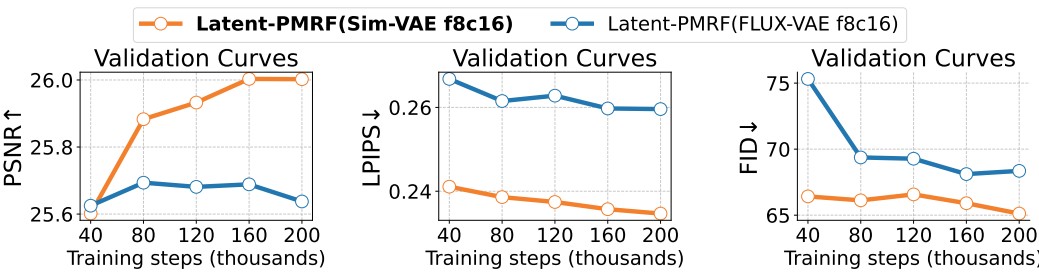

Figure 9: Convergence comparison of VAEs.

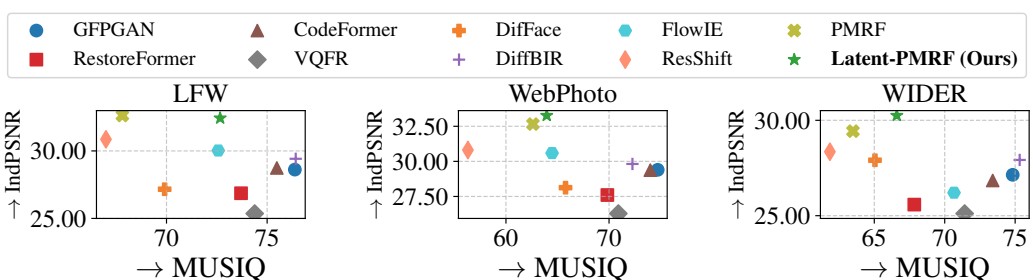

Figure 10: Comparisons on real-world datasets.

## D    STATEMENT ON THE USE OF LARGE LANGUAGE MODELS (LLMS)

In adherence to the ICLR 2026 submission guidelines, this section details the use of a Large Language Model (LLM) assistant during the preparation of this manuscript. The LLM, acting as a research and writing co-pilot, played a significant role in refining the manuscript's structure, language, and presentation. The authors maintained full intellectual control throughout the process and take complete responsibility for all content.

The precise role of the LLM can be categorized as follows:

- **Manuscript Writing and Polishing:** The authors wrote the initial drafts for all sections of the paper, providing the key technical details, experimental results, and core arguments. The LLM was then used extensively as an interactive writing assistant to:
  - **Enhance Clarity and Conciseness:** Rephrasing long or complex sentences to improve readability and flow.

- **Improve Academic Tone:** Suggesting more formal and professional vocabulary and sentence structures appropriate for a top-tier conference submission.
- **Correct Grammar and Syntax:** Performing comprehensive proofreading to identify and correct grammatical errors, typos, and awkward phrasing.
- **Suggest Alternative Phrasing:** Providing multiple options for expressing a single idea to avoid repetitive language).

- **Technical Formalization:** For complex mathematical sections, such as the derivation of the Theorem 1,2, the authors provided the core mathematical steps and overall idea. The LLM assisted in translating these steps into a clear, well-structured narrative and formatting them professionally in LaTeX. The LLM did not generate novel mathematical proofs but rather helped in their presentation.

Throughout this collaborative process, every suggestion and piece of text generated by the LLM was critically reviewed, edited, and approved by the human authors. The authors are solely responsible for the scientific validity, originality, and all claims made in this paper. The LLM is not considered an author.

