# OpenReview forum: "Latent Posterior-Mean Rectified Flow for Higher-Fidelity Perceptual Face Restoration"
_ICLR.cc/2026/Conference — ICLR 2026 Conference Withdrawn Submission_

### Official Review · Reviewer_8Pix · 2025-10-25

**Soundness:** 2
**Presentation:** 3
**Contribution:** 2
**Rating:** 4
**Confidence:** 5

**Summary:**

This paper addresses the problem of face image restoration, a highly practical task with numerous applications. To solve this problem, the authors propose Latent Posterior-Mean Rectified Flow (Latent-PMRF), a reformulation of Posterior-Mean Rectified Flow (PMRF) in the latent space of a variational autoencoder. Specifically, PMRF is an image restoration algorithm that attempts to perform optimal transport between the posterior mean (MMSE estimator) distribution and the ground-truth image distribution, which yields the theoretically optimal solution of the perception-distortion tradeoff under a perfect perceptual quality constraint (to optimally restore an image, first compute the posterior mean, and then apply the optimal transport map). Latent PMRF does exactly the same, but in the latent space of a VAE. Namely, it learns to transport the latent distribution of the posterior mean estimates to the latent distribution of the ground-truth images.
To my understanding, the main advantage of Latent-PMRF is its significantly faster training time compared to PMRF. The experiments, which are quite extensive, show competitive performance with PMRF.

**Strengths:**

1. The authors propose a new architecture for VAEs, which yields much lower distortion when encoding-decoding face images, compared to the VAEs that are typically used by diffusion models.
2. The proposed Latent-PMRF can achieve similar performance to PMRF, yet it requires a much lower training time. This is reasonable since the authors pre-train a VAE.
3. The paper is overall well written and very easy to follow.
4. The proposed solution is somewhat original: most diffusion-based image restoration algorithms attempt to sample from the posterior. The proposed solution is an approximate optimal transport in the latent space of a VAE, inspired by the fact that optimal transport in pixel space is ideal in terms of the perception-distortion tradeoff (which is the motivation for the PMRF paper).

**Weaknesses:**

1. While Latent-PMRF requires lower training times, it still relies on a VAE tailored for face images, which by itself requires training. If the sole purpose of the VAE is to serve the Latent-PMRF model (because the trained Sim-VAE is designed for face images), then it's not immediately clear whether Latent-PMRF requires less training -- the authors must also report the training times of Sim-VAE, and add them to those of Latent-PMRF. However, if Latent-PMRF can also work well for general-purpose VAEs (the appendix shows that it doesn't), then the current story can hold: the training times of Latent-PMRF are the only ones that matter.
2. The assumptions in the theory of the paper are too strong and misleading. Specifically, to prove the theorem, the authors assume that the VAE is an isometry, which means that it preserves the Euclidean distance between any pair of inputs. But this is rarely the case, as an isometric VAE is useless. Indeed, if the VAE is an isometry, then the latent space is essentially equivalent to the pixel space, which makes the VAE not useful at all (for semantic manipulations, dimensionality reduction, etc.).
3. Solving image restoration problems through approximate optimal transport in the latent space of a VAE has been proposed before, see [1]. I am aware that the proposed Latent-PMRF solution is significantly different than [1], but I still think it's important to discuss the differences.

Last note: the authors refer to [2] as concurrent work, but [2] is a preprint that was uploaded to arXiv in February 2025. Since it has not been peer-reviewed, I believe the current submission and the proposed solution can be considered original work.

[1] Adrai et al. Deep Optimal Transport: A Practical Algorithm for Photo-realistic Image Restoration. NeurIPS 2023.
[2] Cohen et al. Efficient Image Restoration via Latent Consistency Flow Matching. arXiv 2025, 2502.03500.

**Questions:**

1. What are the training times of Sim-VAE? Is the combined training time with Latent-PMRF still much smaller than PMRF?
2. To what extent is the Sim-VAE an isometry? Notice that a VAE can have very good encoding-decoding distortion, but this does not imply that the encoder is an isometry. I think this is an important aspect that is left out of the paper, because it serves a central role in determining whether it makes sense to solve an L2 optimal transport problem in latent space.

---

### Official Review · Reviewer_fQ6L · 2025-10-31

**Soundness:** 2
**Presentation:** 2
**Contribution:** 2
**Rating:** 4
**Confidence:** 5

**Summary:**

This work introduces a framework called Latent-PMRF for face restoration. The core contributions include reformulating Posterior-Mean Rectified Flow (PMRF) in the latent space of a Variational Autoencoder (VAE), which improves alignment with human perception and bounds the minimum distortion by the VAE’s reconstruction error. The authors also propose Sim-VAE, an improved VAE architecture that significantly boosts reconstruction and restoration performance. Latent-PMRF achieves a 5.79× speedup over PMRF in terms of FID and demonstrates a superior perception-distortion trade-off in blind face restoration, producing high-quality and consistent results.

**Strengths:**

1. The paper proposes Latent-PMRF, which implements Posterior-Mean Rectified Flow (PMRF) in VAE latent space, motivated by the better alignment of feature-space distances with human perception compared to pixel space.
2. It distinguishes between two latent-space source distributions—latent of the posterior mean vs. posterior mean of the latent—and shows that the former preserves the minimum-distortion property, with theoretical support.
3. The authors introduce Sim-VAE, a streamlined variant of SD-VAE with architectural improvements (e.g., simplified ResBlocks, pixel-wise LayerNorm, no mid-layer self-attention), which empirically enhances both reconstruction and restoration performance over existing VAEs.

**Weaknesses:**

1. The inference speed of Latent-PMRF is slightly slower than PMRF due to VAE encoding/decoding, but the tradeoff between this overhead and perceptual gains is not quantitatively analyzed.
2. While Sim-VAE improves performance, the ablation study lacks qualitative analysis (e.g., feature map visualization) to explain why certain architectural changes—like replacing GroupNorm with LayerNorm—are beneficial.
3. The comparison with concurrent work ELIR is indirect; a direct experimental comparison would better justify the claimed advantage of using “latent of posterior mean” as the source distribution.
4. Training details for Sim-VAE (e.g., patch size choice, KL weight tuning) are briefly mentioned, which may affect reproducibility.

**Questions:**

1. Could the VAE and rectified flow be trained jointly to further reduce the total distortion, rather than using a fixed VAE?
2. Is Sim-VAE’s design specific to face restoration, or would it also benefit general image restoration tasks like deblurring or super-resolution?
3. How sensitive is Latent-PMRF to the quality of the posterior mean estimator? Would a weaker estimator significantly degrade performance?
4. In Table 3, increasing latent channels improves reconstruction PSNR but slightly worsens restoration LPIPS—does this suggest a tradeoff between fidelity and perceptual enhancement?

---

### Official Review · Reviewer_viEq · 2025-10-31

**Soundness:** 3
**Presentation:** 3
**Contribution:** 3
**Rating:** 6
**Confidence:** 2

**Summary:**

This paper proposes Latent-PMRF, a latent-space reformulation of Posterior-Mean Rectified Flow (PMRF) for high-fidelity perceptual face restoration, addressing the limitation of PMRF’s pixel-space modeling that poorly aligns with human perception. By conducting rectified flow in the latent space of a variational autoencoder (VAE), Latent-PMRF optimizes perceptual quality more efficiently while retaining PMRF’s minimal distortion property. The source distribution is defined as the latent representation of the posterior mean (minimum-distortion estimate), bounding the minimum distortion by the VAE’s reconstruction error. To enhance performance, the authors introduce Sim-VAE, a simplified VAE with architectural improvements (e.g., simplified ResBlocks, pixel-wise layer normalization, removed middle-layer self-attention) and optimized training loss (excluding adversarial loss). Experiments on CelebA-Test and real-world datasets (LFW, WebPhoto, WIDER) show Latent-PMRF achieves a superior perception-distortion (PD) tradeoff, a 5.79× FID convergence speedup over PMRF, and outperforms state-of-the-art methods in key metrics like PSNR, LPIPS, and identity preservation (Deg, LMD).

**Strengths:**

1. Latent-PMRF leverages the VAE's latent space, enabling more effective optimization aligned with human perception. This results in better perceptual quality and faster convergence compared to PMRF in pixel space.

2. The reformulation of PMRF in latent space, combined with the use of a Sim-VAE that enhances VAE reconstruction fidelity, provides a significant boost to restoration performance, addressing challenges in face restoration tasks effectively.

3. The method accelerates the training process by 5.79x in terms of FID, enabling faster and more computationally efficient training, making it highly effective in resource-constrained environments.

**Weaknesses:**

1. The removal of self-attention in Sim-VAE’s middle layers, while improving resolution generalization, may sacrifice the model’s ability to capture long-range contextual dependencies in facial images, limiting performance on complex facial structures or textures.
2. Sim-VAE’s training relies on a fixed combination of LSDIR and FFHQ datasets, lacking validation on diverse data distributions (e.g., low-light, extreme poses) which may hinder its generalization to real-world edge cases.
3. Despite the speed improvements in flow optimization, the encoding and decoding processes of the VAE remain relatively slow. This could reduce the overall inference speed compared to pixel-space methods like PMRF, which may be a disadvantage in real-time applications.
4. The method is primarily designed for blind face restoration, raising questions about its applicability to other image restoration tasks, especially those involving different structural or domain-specific features.

**Questions:**

See weaknesses

---

### Official Review · Reviewer_ZySu · 2025-11-01

**Soundness:** 3
**Presentation:** 2
**Contribution:** 2
**Rating:** 4
**Confidence:** 3

**Summary:**

This work proposes Latent-PMRF, a novel method that effectively advances the perception-distortion (PD) tradeoff. It identifies a key limitation in the prior Posterior-Mean Rectified Flow (PMRF) approach—its pixel-space modeling misalignment with human perception. To address this, Latent-PMRF innovatively reformulates the PMRF framework within a learned latent space, which more closely aligns with perceptual evaluation. A significant finding is the critical role of the VAE's design, leading to the proposed Sim-VAE that excels in both reconstruction and restoration tasks. Extensive experiments in blind face restoration validate the method's superiority, demonstrating not only an improved PD trade-off but also a remarkable 5.79× convergence speedup over PMRF. This work provides a valuable paradigm for efficient, high-perceptual-quality image restoration.

**Strengths:**

+ Enhanced Perceptual Alignment and Efficiency: By reformulating the PMRF model in a learned latent space rather than pixel space, the method achieves superior alignment with human perception, leading to a significant 5.79× speedup in convergence as measured by FID.
+ Theoretically-Bounded Distortion with a Superior VAE: The novel design of the source distribution in Latent-PMRF guarantees that the minimum achievable distortion is bounded by the VAE's reconstruction error. This advantage is further amplified by the proposed Sim-VAE, which is specifically optimized for restoration and significantly boosts final performance.
+ Improved Overall Perception-Distortion Trade-off: Extensive experiments validate that the method achieves a better overall balance between perceptual quality and fidelity (the PD trade-off), producing visually appealing results that maintain high consistency with the input images.

**Weaknesses:**

- There are no ablation studies to verify the effect of the Latent-PMRF and Sim-VAE， respectively.
- The organization is ugly. In the abstract, I do know what the meaning of the Sim-VAE.
-  When compared with other methods, some performance metrics are not always the best. The authors do not provide detailed explanations. Additionally, the runtimes and the number of parameters do not outperform existing methods.
- The datasets to be verified are simple. There is a simple face in one image. In addition, the ground truth of the real-world dataset is unclear. The qualitative evaluation results on the real-world dataset are unbelievable.
 - The authors do not present the limitations of the proposed method.

**Questions:**

Please see the weaknesses.

---

### Note · Authors · 2025-12-03

**Comment:**

We would like to sincerely thank all reviewers for their valuable feedback and constructive comments. Their insights have highlighted several important directions for improvement. Although we have decided to withdraw the paper from consideration, we will carefully incorporate the reviewers’ suggestions to further strengthen our work. We truly appreciate the time and effort invested in evaluating our submission.

**Withdrawal Confirmation:**

I have read and agree with the venue's withdrawal policy on behalf of myself and my co-authors.